# Phylogeny and Mycotoxin Characterization of Alternaria Species Isolated from Wheat Grown in Tuscany, Italy

**DOI:** 10.3390/toxins10110472

**Published:** 2018-11-14

**Authors:** Francesca A. Ramires, Mario Masiello, Stefania Somma, Alessandra Villani, Antonia Susca, Antonio F. Logrieco, Carlos Luz, Giuseppe Meca, Antonio Moretti

**Affiliations:** 1Institute of Sciences of Food Production, Research National Council (CNR-ISPA), Via Amendola 122/O, 70126 Bari, Italy; francesca.ramires@ispa.cnr.it (F.A.R.); mario.masiello@ispa.cnr.it (M.M.); stefania.somma@ispa.cnr.it (S.S.); alessandra.villani@ispa.cnr.it (A.V.); antonella.susca@ispa.cnr.it (A.S.); antonio.logrieco@ispa.cnr.it (A.F.L.); 2Department of Preventive Medicine, Nutrition and Food Science Area, University of Valencia (Spain), Avenida Vicent Andres Estelles s/n, 46100 Burjassot, Valencia, Spain; carlos.luz@uv.es (C.L.); giuseppe.meca@uv.es (G.M.)

**Keywords:** tenuazonic acid, alternariol, alternariol-monomethyl ether, altenuene, Section *Alternaria*, Section *Infectoriae*, allergen alt1a, glyceraldeyde-3-phosphate dehydrogenase, translation elongation factor 1α

## Abstract

Wheat, the main source of carbohydrates worldwide, can be attacked by a wide number of phytopathogenic fungi, included *Alternaria* species. *Alternaria* species commonly occur on wheat worldwide and produce several mycotoxins such as tenuazonic acid (TA), alternariol (AOH), alternariol-monomethyl ether (AME), and altenuene (ALT), provided of haemato-toxic, genotoxic, and mutagenic activities. The contamination by *Alternaria* species of wheat kernels, collected in Tuscany, Italy, from 2013 to 2016, was evaluated. *Alternaria* contamination was detected in 93 out of 100 field samples, with values ranging between 1 and 73% (mean of 18%). Selected strains were genetically characterized by multi-locus gene sequencing approach through combined sequences of allergen alt1a, glyceraldeyde-3-phosphate dehydrogenase, and translation elongation factor 1α genes. Two well defined groups were generated; namely sections *Alternaria* and *Infectoriae*. Representative strains were analyzed for mycotoxin production. A different mycotoxin profile between the sections was shown. Of the 54 strains analyzed for mycotoxins, all strains included in Section *Alternaria* produced AOH and AME, 40 strains (99%) produced TA, and 26 strains (63%) produced ALT. On the other hand, only a very low capability to produce both AOH and AME was recorded among the Section *Infectoriae* strains. These data show that a potential mycotoxin risk related to the consumption of *Alternaria* contaminated wheat is high.

## 1. Introduction

Since ancient times, wheat has been one of the most cultivated cereals worldwide, representing an important source of carbohydrates in human and livestock diet. Italy is one of the most important producers at worldwide level with production of around 3 million of tons of soft wheat and 4.5 million of tons of durum wheat in 2018 [1].

Under specific pedo-climatic conditions, wheat plants can be colonized by several fungal species since the early growth stages, although the flowering stage is the most susceptible period for fungal colonization. As a consequence, kernels can be colonized by several toxigenic fungal species, mainly belonging to *Fusarium* and *Alternaria* genera. These fungal genera negatively influence quantitative and qualitative wheat production and they represent a serious toxicological risk as they produce a broad spectrum of mycotoxins and secondary metabolites, which can cause problems in humans and animals [2] (pp. 51–57).

Several *Alternaria* species colonize wheat plants, causing symptoms on both leaf and kernels. In particular, some *Alternaria* species, such as *A. triticina*, colonize wheat leaf causing leaf spot disease [3,4,5]. On the other hand, *A. alternata*, *A. arborescens*, *A. infectoria*, *A. tenuissima*, and *A. triticina* are common colonizers of wheat kernels, being associated with black point disease at worldwide level [6,7,8].

Wheat kernels infected by *Alternaria* species are characterized by black pigmentation in the underlying embryo region, which causes a reduction in flour quality for the release of black points in the flour and in the resulting bread baked, and the loss of nutritional value with a decrease of starch and soluble carbohydrates, and increase of phenolics, prolamin, and gluten content [9]. In addition, the food consumption of *Alternaria* contaminated wheat may be a major risk to human health, as many *Alternaria* species produce several metabolites, which are toxic for humans and animals [6,10,11].

The most important *Alternaria* mycotoxins are the dibenzopyrone derivatives alternariol (AOH), alternariolmonomethyl ether (AME), and altenuene (ALT); the tetramic acid derivatives tenuazoic acid (TA); the cyclic tetrapeptidetentoxin (TEN); the perylene derivatives alterotoxins (ATX-I, ATX-II, ATX-III); and the sphinganine-analog *Alternaria alternata f. sp. lycopersici* toxin (AAL toxin) [6,12,13,14].

Several studies demonstrate the toxicological risk linked to these metabolites. The mycotoxins AOH and AME show genotoxic activity on human cell lines of colon cancer [11]. The consumption of wheat highly contaminated by AOH and AME was significantly related to elevated levels of human esophageal cancer in China [15]. Tenuazoic acid has been associated with human hematologic disorder called “onalay”, in central and southern Africa [16]. Moreover, in another study, this myicotoxin was observed to induce precancerous changes in the esophageal mucosa of mice [17]. Altertoxin I showed high acute toxicity in mice and mutagenicity of mammalian cells lines [18]; moreover, ATX I and ATX-III were shown to be potent mutagens and tumor promoters [11]. Finally, the sphinganine-analog AAL-toxin is harmful to humans, causing apoptosis of mammalian cells interfering with ceramide biosynthesis [10]. Altogether, *Alternaria* mycotoxins have been associated with colon-rectal cancer, occurring significantly in food consummated by humans affected by this pathology [19]. For this reason, in recent decades, food and feed contamination by mycotoxins produced by *Alternaria* species is a main issue, as the European Food Safety Authority (EFSA) has highlighted [2]. Deeper studies about toxic effects of these metabolites and monitoring activities to detect the presence of *Alternaria* toxins in food and feed are encouraged in order to define and evaluate risk exposure in European Union countries.

Among *Alternaria* species occurring on wheat kernels, a species-specific mycotoxin profile has not been identified. However, the species *A. alternata*, *A. tenuissima*, and *A. arborescens* have been reported able to produce AOH, AME, ATX-I, II, III, TA, and TEN [6,12,13,14]. On the other hand, *A. infectoria* species seem unable to produce mycotoxins [6]. Therefore, a correct identification of *Alternaria* strains occurring on wheat kernels is needed in order to better define the potential risk of their accumulation in the final products. However, such identification at species level is very difficult in this genus because of their highly controversial taxonomic “status” [8]. The first approach widely used in *Alternaria* species identification has been for several years based on the observation of phenotypical characters as sporulation pattern, total length of the primary conidiophores, type and origin of branching, conidial shape, and orientation [20]. Subsequently, the DNA-based species identification is becoming prevalent, by analyzing DNA fingerprinting (RAPD, PCR-RFLP, AFLP, and ISSR) or DNA and protein-coding gene sequences, such as *glyceraldehyde 3-phosphate dehydrogenase*, *endopolygalacturonase*, and *β-tubulin*, *allergen Alt 1a* [21,22,23,24]. The molecular approach based on gene sequencing allowed the promotion of important taxonomic revisions of *Alternaria* genus grouping species in sections [25,26]. *Alternaria* species are characterized by a great morphological, molecular, and mycotoxin variability. Therefore, a polyphasic approach based on morphological, mycotoxin profile, and molecular characterization has been suggested for a correct identification [27].

In this paper, we investigated and characterized an *Alternaria* set of strains from wheat in Tuscany, to monitor it in the same geographical area along four consecutive crop seasons, from 2013 to 2016. The aims were as follows: (a) to evaluate the occurrence and the dynamics of *Alternaria* contamination on wheat in Tuscany; (b) to morphologically and molecularly identify species within the isolated *Alternaria* population; and (c) to analyze the mycotoxin profile of representative strains to assess the correct *Alternaria* mycotoxin risk due to consumption of wheat grown in Tuscany.

## 2. Results

### 2.1. Detection of Fungal Contamination

For each sample, after five days of incubation, fungal colonies originating from infected kernels were counted and percentages of total fungal contamination and *Alternaria* contamination were calculated. Mean values of contamination, obtained considering host and crop seasons, are reported in Table 1. With the exception of two samples of durum wheat and one sample of soft wheat, all samples were highly contaminated by fungal species with values of fungal contamination ranging between 47 and 100%. Besides *Alternaria*, the main occurring fungal genera were *Fusarium* and, to a lesser extent, *Cladosporium*, *Epicoccum*, and *Stemphylium* (Appendix A).

*Alternaria* contamination was detected in 93 out of 100 samples. In 13 samples, *Alternaria* contamination values were below 10%, and in the remaining 80 samples, the *Alternaria* occurrence ranged from 11% to 73% of contaminated kernels. The highest *Alternaria* contamination was detected in two fields located in Firenze province, during 2015, with values of 54 and 73%, respectively. Overall, in soft wheat, the highest *Alternaria* contamination was detected in samples collected in Firenze (35%) and Siena (36%) provinces during 2016 crop season, while in durum wheat, the highest was detected in samples collected in Siena province during 2015 (43%). The lowest *Alternaria* contamination was detected during the 2014 crop season, in all areas sampled, both in soft and durum wheat, with values ranging between 0 and 17% (mean values of 3.6) and between 0 and 11% (mean values of 6.3%), respectively (Table 1).

Mean values of temperatures and rainfall, calculated for each crop season (2013–2016) and from heading to harvest of each crop season, are reported in Table 2. The highest rainfall was recovered in 2013, with 820 mm over the crop season. Moreover, the highest rainfall was associated with a lower mean temperature (9.1 °C during the whole crop season and 14.8 °C from heading to harvest). The highest temperature was registered during the 2016 crop season (12.7 °C). Considering the period from heading to harvest, in which wheat plants are most susceptible to fungal infections, the highest rainfall (256 mm) and temperature (17.6 °C) values were observed in 2016. In the same years, the highest contamination by *Alternaria* species was also observed (mean value of 32.8%). On the contrary, in the same period, from heading to harvest, the lowest levels of rain and temperature were detected in 2014 (95 mm of rain and 15 °C, respectively), associated with the lowest *Alternaria* contamination (mean value of 5.5%).

From each sample, *Alternaria* strains were morphologically characterized. A great variability was observed based on the shape of conidia, and conidiophore branches’ morphology. All strains were related to four morphospecies: *A. alternata*, *A. tenuissima*, *A. arborescens,* and *A. infectoria* [20]. Representative strains were thus subjected to gene sequencing for confirming species identification and establishing phylogenetic relationships within the investigated population.

### 2.2. DNA-Based Identification

The evolutionary history of 134 *Alternaria* strains was studied at the genetic level by amplifying three different fragments of *gpd-*, *alt-*, and *tef1-a*genes. The amplifications generated fragments of about 600, 400, and 590 nucleotides, respectively. All three gene fragments showed length polymorphisms due to intraspecific variation. To further solve the identity of the strains, the phylogenetic analysis of the concatenated sequences of the three fragments was carried out. The phylogenetic tree, obtained with Mega7 software using the maximum parsimony method, allowed us to define five well-separated clades, corresponding to Section *Alternaria* (A), Section *Brassicicola* (B), Section *Porri* (C), Section *Infectoriae* (D), and Section *Pseudoalternaria* (E), as determined using reference strains (Figure 1). The resolution of all the clades was supported by high bootstrap values. In Sections *Brassicicola* and *Porri*, only reference strains included in the analysis were clustered (Figure 1). In clade A (Section *Alternaria*), six well-defined sub-clades were obtained. Sub-clade A1 grouped 70 field strains with the reference strains *A. alternata* ATCC66891, *A. alternata* ATCC 11680, *A. alternata* BMP 0270, *A. citrimacularis* BC2-RLR-17s, *A. angustiovoidea* EGS 36-172, *A. tenuissima* BMP 0304, *A. limoniasperae* BMP 2335, *A. perangusta* BMP 2336, and *A. turkisafria* BMP 3436. In this sub-clade, an elevated level of homology was observed. Sub-clade A2 grouped 11 field strains with *A. arborescens* BMP 0308 and *A. cerealis* EGS 43-072 reference strains. In sub-clade A3 and sub-clade A4, only reference strains were grouped: *A. fragariae* BMP 3062, *A. gaisen* BMP 0243, and *A. gaisen* BMP 2338 in sub-clade A3; *A. longipes* BMP 0313, *A. grisea* CBS 107.36, *A. tangelonis* BMP 2327, and *A. burnsii* CBS 107.38 in sub-clade A4 (Figure 1). Sub-clade A5 grouped 24 field strains and the reference strains *A. citriarbusti* BMP 2343, *A. mali* BMP 3064, and *A. rhadina* CBS 595.93. Sub-clade A6 included only *A. betae-kenyensis* CBS 118810 reference strains. Clade D grouped 17 field strains and all reference strains of *Alternaria* species belonging to “Section *Infectoriae*”: *A. alternarina* CBS 119396, *A. intercepta* EGS 49-137, *A. incomplexa* EGS 17-103, *A. oregonensis* EGS 29-194, *A. metachromatica* EGS 38-132, *A. californica* EGS 52-082, *A. hordeicola* EGS 50-184, *A. conjuncta* EGS 37-139, *A. ethzedia* EGS 37-143, *A. infectoria* EGS 27.193, *A. photistica* EGS 35-172, *A. viburni* EGS 49-147, *A. novae-zelandiae* EGS 48-092, *A. ventricosa* EGS 52-075, and *A. triticina* EGS 17-061, in addition to 17 field strains. In particular, ITEM 17966 showed high homology with *A. ethzedia* EGS 37-143; ITEM 17974 showed high homology with *A. metachromatica* EGS 38-132; six field strains were highly similar to *A. ventricosa* EGS 52-075; and nine field strains were highly similar to *A. triticina* EGS 17-061. Clade E “Section *Pseudoalternaria*” grouped four field strains and *A. rosae* EGS 41-130 reference strains (Figure 1). More detailed information is reported in Appendix A.

### 2.3. Mycotoxin Production Profile

Fifty-four *Alternaria* strains were analyzed for mycotoxin profile. For each strain, AOH, AME, ALT, and TA productions were evaluated (Table 3). Chemical analyses were carried out on 41 strains included in Section *Alternaria* (Clade A: 28, 3, and 10 strains grouped in sub-clade A1, A2, and A5, respectively); 12 strains included in Section *Infectoriae* (Clade D); and 1 strain included in Section *Pseudoalternaria* (Clade E). With regard to AOH production, all *Alternaria* strains grouped in Section *Alternaria* (Clade A) produced this mycotoxin, in a range from 0.5 to 5620 mg·kg^−1^. In Table 3, the values for each strain analyzed are reported. In particular, AOH was produced at maximum values of 5620, 3180, and 3650, mg·kg^−1^ for sub-clade A1, A2, and A5, respectively. Overall, AOH amount ranged in an interquartile between 57.1 and 1083.0 mg·kg^−1^, with a median value of 342.3 mg·kg^−1^ (Figure 2). On the other hand, all strains grouped in Section *Infectoriae* produced AOH at a very low level (mean value 4 mg·kg^−1^; range 0.3–20 mg·kg^−1^). In particular, AOH amount ranged in an interquartile between 0.7 and 9.1 mg·kg^−1^, with a median value of 2.6 mg·kg^−1^ (Figure 2). Also, the only Section *Pseudoalternaria* strain analyzed, ITEM 17904 (Figure 1), produced a low amount of AOH (1.5 mg·kg^−1^).

The production of AME has a similar trend to that of AOH. All strains grouped in Section *Alternaria* produced AME with mean values of 3066, 678, and 2676 mg·kg^−1^ for A1, A2, and A5 sub-clades, respectively (Table 3). In Section *Alternaria*, AME amount ranged in an interquartile between 24.9 and 2392.0 mg·kg^−1^ (median value of 776.4 mg·kg^−1^; Figure 2). All strains of the Section *Infectoriae* produced a very low amount of this mycotoxin (interquartile ranging between 1.4 and 18.0 mg·kg^−1^ with a median value of 6 mg·kg^−1^). The strain analyzed belonging to Section *Pseudoalternaria* produced also a very low amount of AME: 5.5 mg·kg^−1^ (Table 3).

The production of ALT was low for both Sections *Alternaria* and *Infectoriae*. In Section *Alternaria*, 10 out of 28 strains grouped in sub-clade A1, 2 out of 3 strains grouped in sub-clade A2, and 3 out of 10 strains grouped in sub-clade A5 did not produce ALT. Moreover, the ALT amount did not exceed values of 180, 0.5, and 130 mg·kg^−1^ for A1, A2, and A5, respectively (Table 3). In Section *Alternaria*, ALT amount ranged in an interquartile between 0 and 12.6 mg·kg^−1^, with a median value of 1.2 mg·kg^−1^ (Figure 2). In Section *Infectoriae* (interquartile ranging between 0.0 and 6.8 mg·kg^−1^ with a median value of 0.1 mg·kg^−1^, Figure 2), only four strains out of twelve produced ALT with a range of 0.2–9.6 mg·kg^−1^ and a mean of value of 1.7 mg·kg^−1^. Finally, the Section *Pseudoalternaria* strain produced 10 mg·kg^−1^ of ALT (Table 3).

With regard to TA production, all strains included in Section *Alternaria*, with the exception of ITEM 17880 (sub-clade A2), were able to produce the mycotoxin in an interquartile range between 17.9 and 180.8 mg·kg^−1^, with a median value of 68.6 mg·kg^−1^ (Figure 2). On the other hand, in Section *Infectoriae* (interquartile ranging between 0.0 and 75.4 mg·kg^−1^; median value of 11.4 mg·kg^−1^, Figure 2), a great variability in TA production was observed. Six out of twelve strains did not produce TA, while the six producer stains ranged from 4 to 415 mg·kg^−1^ (Table 3). The strain of Section *Pseudoalternaria* analyzed did not produce TA.

## 3. Discussion

This paper focused on the characterization of an *Alternaria* population isolated from wheat, exploiting, in four consecutive crop seasons (2013–2016), an extensive monitoring activity of both durum and soft wheat samples collected in Tuscany, central Italy, a particularly suitable region for wheat production. Italy is among the most important producers of wheat and wheat-derived food worldwide, and much attention is addressed to quality agricultural productions being characterized as the “Made in Italy” foodstuffs. Since decades ago, several studies have been carried out to monitor mycotoxigenic fungal species and mycotoxin contamination on wheat and other cereals in Italy [28,29]. However, almost all studies were focused on *Fusarium* and related mycotoxins, while *Alternaria* occurrence has been poorly investigated. Black point disease has been considered for a long time as a qualitative and commercial problem, linked to both release of black points in flour and baked bread, as well as loss of nutritional value [9]. Several studies have demonstrated the toxicological risks of *Alternaria* mycotoxins on both human and animals [11,30,31]. Furthermore, EFSA (European Food Safety Agency) triggered the need for deeper studies on *Alternaria* species and related mycotoxin occurrence risk on foodstuffs [2], while specific regulation for *Alternaria* mycotoxins is under evaluation at European Union level. The samples analyzed in this study were highly contaminated by fungal species belonging mainly to *Alternaria*, *Cladosporium*, *Epicoccum*, *Fusarium*, and *Stemphylum* genera (97 samples out of 100). *Alternaria* occurrence was very high in terms of incidence and frequency. Indeed, 80 out of the 100 samples analyzed were contaminated by *Alternaria* at a value higher than 10% of kernels. This high level of contamination highlights the wide risk of *Alternaria* mycotoxin occurrence, as many of the species identified are able to produce mycotoxins. However, the taxonomy of the *Alternaria* genus is a very controversial issue. Indeed, many studies have been devoted to *Alternaria* and related genera, often confirming a confused taxonomic background, which has caused a continuous process of taxonomic revision. In our work, the strains were phylogenetically identified as belonging mostly to Section *Alternaria* and, to a lesser extent, to Section *Infectoriae*, while the occurrence of Section *Pseudoalternaria* strains was rare. Therefore, because of the presence of several toxigenic species among those belonging to Section *Alternaria*, a correct identification of most occurring species on wheat is not only useful to define their identity, but is also of key importance to assess the toxicological risk caused by the *Alternaria* species associated to wheat.

The *Alternaria* strains collected were morphologically identified according to Simmons [20] as *A. alternata*, *A. tenuissima*, *A. arborescens*, and *A. infectoria*. Several studies reported *A. alternata*, *A. tenuissima*, and *A. infectoria* as the most frequent *Alternaria* species isolated from wheat-infected kernels [12,32,33]. *Alternaria arborescens* has been also identified in wheat, as reported in recent studies in Argentina [12,34], Tunisia [35], Germany, and Russia [33]. However, an identification based only on morphological characters can bring misidentification. Thus, for a more reliable identification at species level, we applied a polyphasic approach to our set of *Alternaria* strains, based not only on morphology, but also on genetic and mycotoxin characterization [8,23,25,36]. The three genes used for a multi-locus sequence approach (*gpd*, *tef*, *alt-1a*) were selected because, from previous studies, they proved to be highly informative in *Alternaria* [8,26,36,37]. All strains were grouped into three sections: *Alternaria*, *Infectoriae*, and *Pseudoalternaria*, according to Woudemberg et al. [26,36] and Lawrence et al. [38]. In Section *Alternaria*, 105 strains were included, with the *A. alternata* and *A. tenuissima* morphospecies being the most widespread species. On the other hand, the phylogenetic tree based on the three genes used in the present work (Figure 1) confirmed that the species *A. alternata*, *A. tenuissima*, *A. citrimacularis*, *A. angustiovoidea*, *A. limoniasperae*, *A. perangusta*, and *A. turkisafria* are genetically indistinguishable, as reported in Woudenberg et al. [26,36] and in Somma et al. [8]. Thus, we support the suggestion of the above-mentioned authors to synonymize these species in the single species *A. alternata*.

Twenty-seven *Alternaria* strains were grouped in Section *Infectoriae*, sharing a high level of similarity. In particular, almost all the strains (24 strains) clustered with the reference strains *A. triticina* EGS17-061 and *A. ventricosa* EGS52-075. *Alternaria triticina* is the main *Alternaria* species associated with leaf spot disease of wheat plants, rarely reported as species capable to cause infection of wheat kernels [7]. Moreover, the strain ITEM 17974 clustered with *A. metachromatica* and *A. californica* reference strains; the strain ITEM 17966 was identified as *A. ethzedia*; the strain ITEM 17877 showed a very high level of similarity to *A. novae-zelandiae*. While *A. metachromatica* and *A. californica* were already reported on wheat [20,39] (p. 602), this is the first study reporting the occurrence on wheat of *A. ethzedia* and *A. novae-zelandiae*, which are common pathogens of *Brassica* plants and carrot, respectively [40,41].

Only three strains (ITEM 17904, ITEM 17931, and ITEM 17936) were identified as belonging to Section *Pseudoalternaria*, the species of which have been reported, by Poursafar et al. [42], as occurring on black point affected cereals only in Iran. This section has been only recently described by Lawrence et al. [25,37], on the behalf of its taxonomic revision of *Alternaria* genus. However, our phylogenetic tree clearly describes this section as being merged in the wider Section *Infectoriae*, thus questioning its separated identity as a section. Indeed, the three above-mentioned strains showed higher genetic similarity with reference strains of *A. triticina* and *A. ventricosa*, than with the reference strains of other Section *Infectoriae* species, such as *A. infectoria*. This is clear evidence that the separation of Section *Pseudoalternaria* within *Alternaria* genus needs more robust genetic support to be accepted.

We investigated whether *Alternaria* population structure was related to crop season and host species, but no differences have been observed. Also, the species identified were all able to colonize both durum and soft wheat, at the same frequency. Only strains genetically identified as *A. mali* showed a significant prevalence on a given host, because they were isolated with an evidently higher frequency from durum wheat than soft wheat (20 strains out of the total 24 strains isolated).

Finally, the representative strains of the different clades, generated from the phylogenetic analysis, were chemically characterized through the determination of their in vitro mycotoxin production by verifying the toxicological risk associated with the occurrence of *Alternaria* strains isolated in kernel samples. The Section *Infectoriae* strains produced low amounts of AOH, AME, ALT, and TA, however, several of the Section *Alternaria* strains produced all tested mycotoxins at very high levels, indicating that a potential risk for a significant contamination of AOH, AME, ALT, and TA in the kernels exists at harvest. This finding is of concern as these mycotoxins have been associated with several diseases in humans and animals [11,15,16,17,18]. In addition, consumers’ exposure and the subsequent exertion of toxic effects is plausible because of the lack of regulation in foods. Moreover, a recent study carried out by Huybrechts et al. [19] has related chronic exposure of low-dose *Alternaria* mycotoxins in food commodities to the onset of colorectal cancer in humans. In the same study, the multi-mycotoxin occurrence was also related to the high risk of colorectal cancer onset. The authors pointed out that based on external exposure assessments, the highest contribution in increasing the cancer onset was related to the consumption of bread and other wheat-derived products.

The high level of *Alternaria* contamination of wheat collected in Tuscany, and the fact that further concern can be derived by the common incidence of other mycotoxins—especially those produced by *Fusarium*—opens questions on their possible impact on antagonistic, additive, or synergistic toxicological interactions. Furthermore, an increased attention on *Alternaria* mycotoxins and their producing species occurrence in wheat, and the need to more carefully evaluate a legislation aimed to regulate their occurrence in food, is required; although, wider and more consistent studies at Italian and worldwide levels are imperative to address decision makers at institutional level.

## 4. Conclusions

The level of *Alternaria* contamination in wheat samples was mostly high along the four years investigated. This study relates for the first time the production of main *Alternaria* mycotoxins, AOH, AME, ALT, and TA, to strains belonging to different species and sections of the *Alternaria* genus, genetically characterized according to the new revised taxonomy proposed by Lawrence et al. [25,37,38] and Woodenberg et al. [36].

The higher occurrence on the wheat kernels of strains belonging to Section *Alternaria* compared with Section *Infectoriae* is thus a reason for further concern because of the dramatic differences in the mycotoxin profiles highlighted above.

## 5. Material and Methods

### 5.1. Origin of Wheat Samples

The one hundred wheat fields analyzed (48 fields of soft wheat and 52 fields of durum wheat) were located in different areas of the Tuscany region, in four consecutive crop seasons: 2013, 2014, 2015, and 2016. Forty-one fields in the first year, 31 fields in the second year, 15 fields in the third years, and 13 fields in the last year were considered (Table 1). Samples of wheat kernels were randomly collected from each field.

Meteorological data were collected, to investigate the eventual influence of climatic conditions on the occurrence of *Alternaria* species in wheat kernels. In particular, rainfall distribution and temperature were measured in each geographical areas in which the fields were located using weather stations. The data were used to obtain mean values of rainfall and temperatures for each crop season. As *Alternaria* species mostly colonize wheat kernels in the last growth stages, mean values of climatic parameters from heading stage to harvest are also reported in Table 2.

### 5.2. Isolation and Morphological Characterization of Alternaria Strains

For each sample, 100 representative kernels were superficially disinfected in a 2% sodium hypochlorite solution for 2 min, washed twice with sterile distilled water for 1 min, and then plated (10 kernels/plate) on potato dextrose agar (PDA) added with pentachloronitrobenzene (PCNB; 500 mg L^−1^), streptomycin (100 mg L^−1^), and neomycin (50 mg L^−1^). After five days of incubation at 25 ± 1 °C under fluorescent light (12 h of photoperiod), for each sample, total fungal contamination and *Alternaria* contamination were detected. Based on morphological traits, representative *Alternaria* strains were selected to obtain mono-conidial cultures. Spores of *Alternaria*, collected by colonies originated from infected kernels, were spread at low density on water agar (WA: 20 g L^−1^ agar Oxoid n. 3) and singly collected after germination, using a dissection microscope, on potato carrot agar (PCA: infusion from 20 g peeled and sliced white potatoes, 20 g carrot kept at 60 °C for 1 h; 15 g L^−1^ agar Oxoid n. 3). According to Simmons [20], after seven days of incubation at 25 ± 1 °C, under an alternating light/darkness cycle of 12 h photoperiod, *Alternaria* isolates were morphologically identified and grouped on the basis of sporulation model and colony morphology.

### 5.3. DNA Extraction

One hundred and thirty-four *Alternaria* mono-conidial strains were cultured for three days on cellophane disks overlaid on PDA plates at 25 °C. Afterwards, mycelium of each strain was collected by scraping, frozen, and lyophilized.

Genomic DNA was extracted and purified from powdered lyophilized mycelia (10–15 mg) using the “Wizard Magnetic DNA Purification System for Food” kit (Promega Corporation, Madison, WI, USA), according to the manufacturer’s protocol. Quantity and integrity of DNA were checked at Thermo-Scientific Nanodrop (LabX, Midland, ON, Canada) and by comparison with a standard DNA (1 kb DNA Ladder, Fermentas GmbH, St. Leon-Rot, Germany) on 0.8% agarose gel after electrophoretic separation.

### 5.4. PCR Amplifications and DNA Sequencing

Allergen alt a 1 (*alt*), glyceraldehyde-3-phosphate dehydrogenase (*gpd*), and translation elongation factor 1α (*tef-1a*) genes were selected for a molecular characterization using a multi-locus sequence approach. For PCR reactions, the following specific primer pairs were used: alt-for/alt-rev [21], gdp1/gdp2 [43], and Alt-tef1/Alt-tef2 [8].

Polymerase chain reaction mixture (15 μL) contained 15 ng of DNA template, 0.45 μL of each primer (10 mM), 0.3 μL of dNTPs (10 mM), and 0.075 μL of Hot Master Taq DNA Polymerase (1 U/μL; 5 Prime). The three fragments were amplified according to PCR conditions reported in Somma et al. [8]. The amplification of *tef* and *alt1a* genes was carried out with the following parameters: one initial denaturation stage at 95 °C for 2 min, followed by 35 or 40 cycles, each consisting of 30 s of denaturation at 95 °C; 30 s of annealing at 58 °C or 55 °C for *alt1a* and *tef* genes, respectively; 50 s of extension at 72 °C; and a final extension stage of 7 min at 72 °C. The thermal cycler parameters for *gpd* gene were as follows: one initial denaturation stage at 95 °C for 2 min, followed by 35 cycles, each consisting of 50 s of denaturation at 95 °C, 50 s of annealing at 59 °C, 60 s of extension at 72 °C, and a final extension stage of 7 min at 72 °C.

The PCR products were visualized with UV after electrophoretic separation in 1X TAE buffer, on 1.5% agarose gel.

PCR products were purified with the enzymatic mixture Exo/FastAP (Exonuclease I, FastAPthermosensitive alkaline phosphatase, Thermo Fisher Scientific (Waltham, MA, USA) and sequenced for both strand with Big Dye Terminator Cycle Sequencing Ready Reaction Kit (Applied Biosystems, Foster City, CA, USA), according to the manufacturer’s recommendations. Both strands were purified by filtration through Sephadex G-50 (5%) (Sigma-Aldrich, Saint Louis, MO, USA) and sequenced in “ABI PRISM 3730 Genetic Analyzer” (Applied Biosystems, Foster City, CA, USA).

### 5.5. Phylogenetic Analyses

For each gene fragment, the FASTA sequences were obtained by using BioNumerics software (Applied Maths, Kortrijk, Belgium). Gene sequences of *Alternaria* species reference strains, listed in Table 4, were downloaded by the National Center for Biotechnology Information (NCBI) and “Alternaria Genomes Database” (AGD). All sequences of the three genes (*alt-a-1*, *gpd*, and *tef1-a*) considered have been aligned using the Clustal W algorithm [44] and the phylogenetic relationships were studied using the maximum parsimony method with MEGA software version 7 [45]. The bootstrap analyses [46] were conducted to determine the confidence of internal nodes using a heuristic search with 1000 replicates, removing gaps. The section designations were made according to Lawrence et al. [37].

### 5.6. Mycotoxins Extraction

The method used for mycotoxin analysis is based on that described by Rubert et al. [47] with some modifications. The samples were finely ground with an Oster Classic grinder (220–240V, 50/60 Hz, 600W; Madrid, Spain). Five grams of each homogenized sample were weighed in a 50 mL plastic tube and 25 mL of methanol was added. The extraction was carried out using an Ultra Ika T18 basic Ultra-turrax, Ika, (Staufen, Germany), for 3 min. The extract was centrifuged at 4000× rpm for 5 min at 5 °C. One milliliter of the supernatant was filtered through a 13 mm/0.22 μm nylon filter and diluted before injection into high performance liquid chromatography associated with a diode array detector (LC-DAD). All the extractions were carried out in triplicate.

#### HPLC Analysis

Alternariol, AME, ALT, and TA were determined using Merk HPLC with a diode array detector (LC-DAD) L-7455 (Merk, Darmstadt, Germany) at 256 nm and Hitachi Software Model D-7000 version 4.0 was used for data analysis. A Gemini C_18_ column (Phenomenex, Torrance, CA, USA) 4.6 × 150 mm, 3 μm particle size was used as the stationary phase. Themobile phase consisted of two eluents, namely eluent A (water with 50 µL/L trifluoroacetic acid) and eluent B (acetonitrile with 50 µL/L trifluoroacetic acid). A gradient program with a constant flow rate of 1 mL/min was used, starting with 90% A and 10% B, reaching 50% B after 15 min and 100% B after 20 min. Then, 100% B was maintained for 1 min. Thereafter, the gradient was returned to 10% B in 1 min and allowed to equilibrate for 3 min before the next analysis [48]. The limit of detection (LOD) and quantification (LOQ) of the method used were of 0.01 and 0.1 ppm, respectively.

The data on the mycotoxin production were statistically processed using the Prism 5 software (La Jolla, CA, USA, www.graphpad.com).

## Figures and Tables

**Figure 1 toxins-10-00472-f001:**
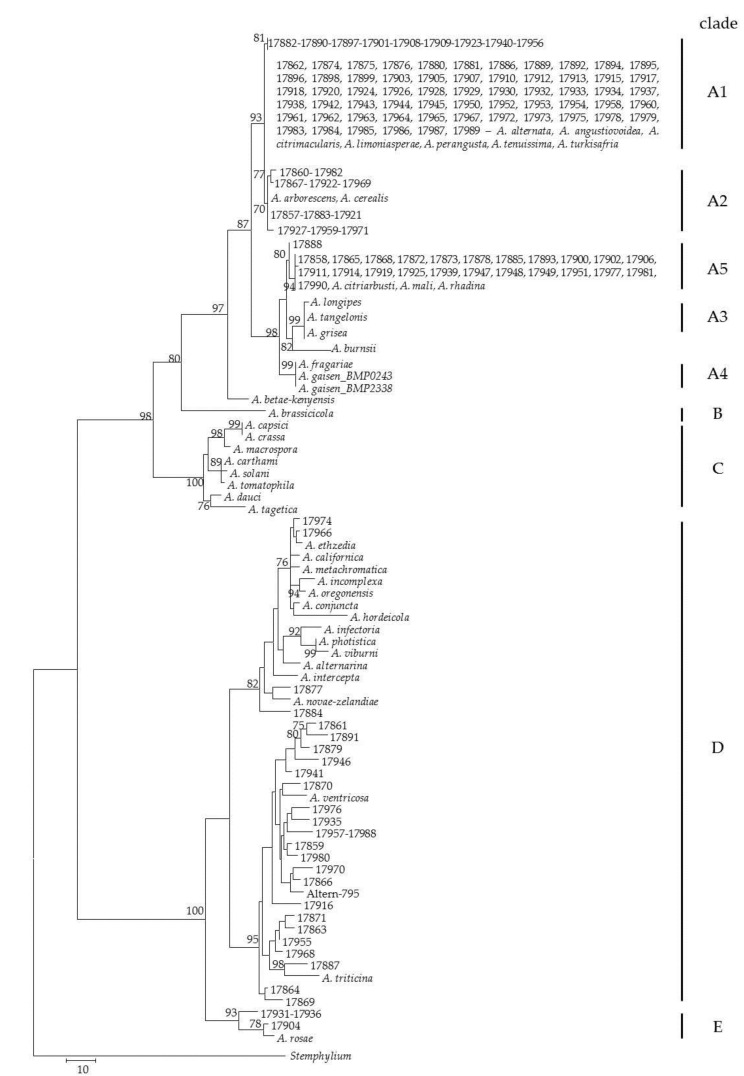
Phylogenetic tree generated by maximum parsimony method (bootstrap 1000 replicates) of combined *alt-*, *gpd-*, and *tef*-gene sequences of 134 *Alternaria* strains.

**Figure 2 toxins-10-00472-f002:**
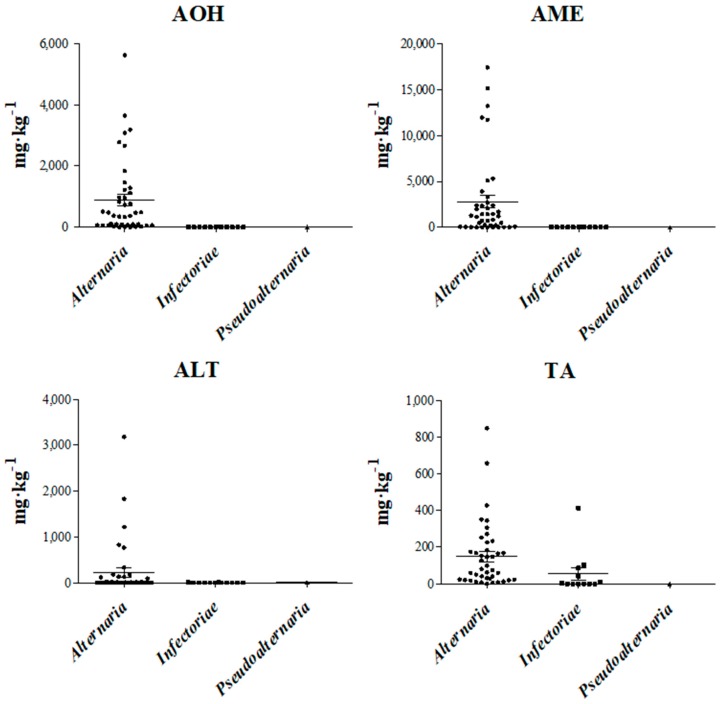
Scatter plot for four examined *Alternaria* mycotoxins: alternariol (AOH), alternariol mono methyl ether (AME), altenuene (ALT), and tenuazonic acid (TA). Production values represent *Alternaria* strains belonging to Section *Alternaria*, *Infectoriae*, and *Pseudoalternaria*. Interquartile and median values are reported in the graphs.

**Table 1 toxins-10-00472-t001:** Mean value and range of *Alternaria* and other fungal contamination detected in the 100 wheat samples collected throughout the Tuscany region, over four consecutive crop seasons (2013–2016).

Host	Year of Sampling	Number of Fields	Fungal Contamination (%)			
			*Alternaria* spp.		Other Fungi	
			Range	Mean Value	Range	Mean Value
Soft wheat	2013	10	0–31	17	36–82	68
	2014	16	0–17	4	19–95	73
	2015	11	4–73	26	15–81	54
	2016	11	11–50	33	36–83	57
Durum wheat	2013	31	0–48	17	1–88	55
	2014	15	0–11	7	57–93	76
	2015	4	8–43	26	36–84	64
	2016	2	22–35	29	60–77	68

**Table 2 toxins-10-00472-t002:** Origin, fungal contamination of wheat samples analyzed, and meteorological data collected in Tuscany.

Year of Sampling	Durum Wheat ^a^	Soft Wheat ^a^	*Alternaria* Contamination (%) ^b^	Meteorological Data (from Seedling to Harvest)		Meteorological Data (from Heading to Harvest)	
				Rainfall (mm)	Mean T	Rainfall (mm)	Mean T
2013	31	10	17.1	820	9.1	181	14.8
2014	15	16	5.5	580	10.9	95	15
2015	4	11	26.2	525	11.1	115	16.6
2016	2	11	32.8	691	12.7	256	17.6

^a^ = number of fields considered; ^b^ = mean value of *Alternaria* contamination for each crop season.

**Table 3 toxins-10-00472-t003:** Mycotoxin production by *Alternaria* strains isolated from wheat in Tuscany.

Strain ^a^	Mycotoxin (mg·kg^−1^) ^b^				Strain ^a^	Mycotoxin (mg·kg^−1^) ^b^			
	AOH	AME	ALT	TA		AOH	AME	ALT	TA
***Alternaria* Section Sub-Clade A1**					***Alternaria* Section Sub-Clade A5**				
17,862	60	710	n.d.	150	17,888	95	1450	n.d.	165
17,874	350	1440	140	305	17,893	335	3	10	17
17,875	475	2400	175	345	17,900	1225	2700	1	255
17,876	960	1700	120	350	17,902	835	15	n.d.	130
17,880	970	15,150	n.d.	n.d.	17,906	0.5	2	3	5.5
17,881	505	2385	15	60	17,865	385	1125	0.5	40
17,882	480	1420	n.d.	235	17,868	1290	3315	n.d	155
17,886	65	480	5	30	17,873	100	724	130	60
17,887	4.5	15	15	20	17,878	20	12.5	3	12.5
17,889	2780	13,230	180	230	17,885	3650	17,415	4	170
17,890	58	475	30	80	Average	794	2676	15	101
17,892	96	20	16	75	Min Value	1	2	0	6
17,894	375	1260	22	42	Max value	3650	17415	130	255
17,895	95	220	n.d.	272	***Infectoriae* Section Clade D**				
17,896	1465	11,710	3.5	848	17,859	1.5	4	n.d.	105
17,897	3090	11,950	9	10	17,861	2.5	23	1.5	4
17,898	110	230	1.0	170	17,863	2.5	1	8.5	n.d.
17,899	1120	2050	2.5	660	17,864	3	3	n.d.	10
17,901	60	75	n.d.	145	17,866	3.5	10	n.d.	n.d.
17,903	730	3910	n.d.	430	17,869	0.5	6	n.d.	n.d.
17,905	70	80	n.d.	23	17,870	0.5	2	n.d.	n.d.
17,907	485	2335	n.d.	175	17,871	10	6	n.d.	n.d.
17,908	75	215	n.d.	20	17,877	2.5	2.5	n.d.	n.d.
17,909	14	7	7.5	25	17,879	0.5	0.5	n.d.	415
17,910	2670	5096	n.d.	185	17,884	20	15	10	40
17,912	25	15	7	50	17,891	2	1	0.2	90
17,858	5620	5290	1.5	65	Average	4	6	2	55
17,911	770	1970	0.5	30	Min Value	0.3	0.5	0	0
Average	842	3066	27	180	Max value	20	23	10	415
Min Value	5	7	0	0	-	***Pseudoalternaria* Section Clade E**			
Max value	5620	15150	180	848	17,904	1.5	5.5	10	n.d.
***Alternaria* Section Sub-Clade A2**									
17,857	3180	1215	0.5	7					
17,867	1835	810	n.d.	7.5					
17,883	2.5	10	n.d.	100					
Average	1673	678	0.2	38					
Min Value	2.5	10	0.0	7					
Max value	3180	1215	0.5	100					

^a^ = code number in ITEM collection; ^b^ = Alternariol (AOH), Alternariol methyl ether (AME), Altenuene (ALT), and Tenuazonic Acid (TA).

**Table 4 toxins-10-00472-t004:** GenBank accession number of *Alternaria* species sequences used in this study.

*Alternaria* Species	Strain Number	GenBank Accession Numbers			
		*Alt-a-1*	*Gpd*	*Tef*	Database *
*A. alternarina*	CBS 119396	JQ905113	JQ905170	JQ905142	NCBI
*A. alternata*	ATCC11680	ATNCTG00005	ATNCTG00056	ATNCTG00621	AGD
*A. alternata*	ATCC66891	AATCTG00058	AATCTG00420	AATCTG00079	AGD
*A. alternata*	BMP 0270	AA2CTG00036	AA2CTG00134	AA2CTG00228	AGD
*A. angustiovoidea*	EGS 36-172	JQ646398	JQ646315	JQ672465	NCBI
*A. arborescens*	BMP 0308	AABCTG00367	AABCTG03225	AABCTG04967	AGD
*A. betae-kenyensis*	CBS 118810	JQ905104	JQ905161	KP125197	NCBI
*A. brassicicola*	EEB 2232	AY563311	AY278813	JQ672450	NCBI
*A. burnsii*	CBS 107.38	JQ646388	JQ646305	KP125198	NCBI
*A. californica*	EGS 52-082	JQ646373	JQ646285	JQ672433	NCBI
*A. capsici*	BMP 0180	AY563298	AY562408	ACSCTG01642	NCBI/AGD
*A. cerealis*	EGS 43-072	JQ646405	JQ646321	JQ672467	NCBI
*A. citriarbusti*	BMP 2343	ACSCTG04746	ACSCTG00332	ACSCTG01642	AGD
*A. citrimacularis*	BC2-RLR-17s	JQ646407	JQ646323	JQ672466	NCBI
*A. conjuncta*	EGS 37-139	AY563281	AY562401	JQ672415	NCBI
*A. crassa*	BMP 0172	AY563293	AY278804	JQ672489	NCBI
*A. dauci*	BMP 0167	AY563292	AY278803	ADCCTG02999	NCBI/AGD
*A. ethzedia*	EGS 37-143	AY563284	AY278795	JQ672427	NCBI
*A. fragariae*	BMP 3062	ACTCTG00345	ACTCTG00074	ACTCTG00439	AGD
*A. gaisen*	BMP 2338	ACRCTG04151	ACRCTG04221	ACRCTG02961	AGD
*A. gaisen*	BMP 0243	JQ646400	JQ646317	JQ672463	NCBI
*A. grisea*	CBS 107.36	JQ646393	JQ646310	JQ672471	NCBI
*A. hordeicola*	EGS 50-184	JQ646372	JQ646284	JQ672425	NCBI
*A. incomplexa*	EGS 17-103	JQ646374	JQ646287	JQ672422	NCBI
*A. infectoria*	EGS 27.193	FJ266502	AY278793	JQ672436.	NCBI
*A. intercepta*	EGS 49-137	JQ646380	JQ646297	JQ672431	NCBI
*A. limoniasperae*	BMP 2335	AFGCTG00301	AFGCTG00774	AFGCTG00104	AGD
*A. longipes*	BMP 0313	ADTCTG24504	ADTCTG20582	ADTCTG20250	AGD
*A. mali*	BMP 3064	AGSCTG02862	AGSCTG00707	AGSCTG00243	AGD
*A. macrospora*	BMP 1949	AMRCTG01538	AMRCTG02006	AMRCTG00463	AGD
*A. metachromatica*	EGS 38-132	AY563285	AY762956	JQ672437	NCBI
*A. novae-zelandiae*	EGS 48-092	JQ646379	JQ646296	JQ672418	NCBI
*A. oregonensis*	EGS 29-194	AY563283	AY762957	JQ672428	NCBI
*A. perangusta*	BMP 2336	JQ646403	JQ646319	JQ672477	NCBI
*A. photistica*	EGS 35-172	AY563282	AY562402	JQ672417	NCBI
*A. rhadina*	CBS 595.93	JQ646399	JQ646316	JQ672470	NCBI
*A. rosae*	EGS 41-130	JQ646370	JQ646279	JQ672414	NCBI
*A.solani*	CBS 116651	AY563299	KC584139	KC584688	NCBI
*A. tangelonis*	BMP 2327	ADCCTG06617	ADCCTG03746	ADCCTG02999	AGD
*A. tagetica*	BMP 0179	AY563297	AY562407	JQ672490	NCBI
*A. tenuissima*	BMP 0304	ALGCTG02124	ALGCTG02071	ALGCTG00260	AGD
*A. tomatophila*	BMP 2032	GQ180101	GQ180085	ATMCTG00738	NCBI/AGD
*A. triticina*	EGS 17-061	JQ646371	JQ646281	JQ672426	NCBI
*A. turkisafria*	BMP 3436	ATKCTG00833	ATKCTG00298	ATKCTG00533	AGD
*A. ventricosa*	EGS 52-075	JQ646377	JQ646290	JQ672426	NCBI
*A. viburni*	EGS 49-147	JQ646375	JQ646288	JQ672420	NCBI

* NCBI = National Center for Biotechnology Information; AGD = Alternaria Genome Database.

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
