# Peer review of "Phylogeny and Mycotoxin Characterization of Alternaria Species Isolated from Wheat Grown in Tuscany, Italy"

_toxins, 2018, doi:10.3390/toxins10110472_

Round 1

Reviewer 1 Report

This paper presents phylogenetics and toxin production data from a surprisingly large and diverse sample of Alternaria sp. strains obtained from wheat fields in Tuscany over 4 years. Phylogenetic analysis was performed according to the most recent classifications available from Lawrence et al. The generally high quality of the science presented in the paper is balanced with a large number of typographic/english errors that make it a times difficult to understand. This paper should be proof read by a native English speaker.

My main remarks are centered on the introduction section and fig.1.

Lines 29-33: I would separate durum and soft wheat, since they are separate species with separate cultivation ranges and uses.

Line 90: I would not use the term "population" when describing a sample of strains beonging to different species.

Figure 1 should be more clear. Font size is really too small, making strain nmes and bootstrap values all bu illegible. In order to reach that, the many strains sharing the exact same sequence than the references strains ATCC66891...BMP0270 should be listed separately. The same can be done for strains identical with BMP3064.

Line 125: “the lowest level of rain and temperature were detected in 2014”: the temperature in 2015 was lower.

There are many typos that I cannot list here, but here were the main errors I found:

l. 35: stage result the -> stage is the

l. 62: micotoxin -> mycotoxin

l. 64 I did not understand “while 63 ATX-II and ATX-III are both cellular transformation includes”

l. 65 results to be -> is

l. 131 morphological -> morphologically.

l. 144: as shown in Figure 1 -> as determined usng reference strains (Figure 1)

l. 147 “subclade   grouped” -> subclade A1 (?) grouped

L 167 and following: mg Kg-1 is not correct. You should use either mg.kg-1 or p.p.m.

l. 218 I did not understand “poses a great attention”

l. 226 I did not understand “solicited the need”

l. 228-232 should be heavily reworded

l. 273: there is no “ref [378]”

l. 285: I did not understand “chemically unraveled”

l.324: “elaborated” -> “analysed” or “used”

Starting l. 338, there are may cases of missing spaces. I detected missing spaces in lines 338, 343, 349, 378, 398, 399, 400, 401, 402, 403.

l. 375: Foe -> For

l. 391: surnatant -> supernatant

l. 395 : AME appears in blue.

Author Response

REVIEWER 1 REPORT

This paper presents phylogenetics and toxin production data from a surprisingly large and diverse sample of Alternaria sp. strains obtained from wheat fields in Tuscany over 4 years. Phylogenetic analysis was performed according to the most recent classifications available from Lawrence et al. The generally high quality of the science presented in the paper is balanced with a large number of typographic/english errors that make it a times difficult to understand. This paper should be proof read by a native English speaker.

My main remarks are centered on the introduction section and fig.1.

Lines 29-33: I would separate durum and soft wheat, since they are separate species with separate cultivation ranges and uses

We decide to report only data from Italy and changed the reference by using data from ISTAT that allowed us to separate between durum and soft wheat production

Line 90: I would not use the term "population" when describing a sample of strains belonging to different species.

Changed in “set of strains”

Figure 1 should be more clear. Font size is really too small, making strain nmes and bootstrap values all bu illegible. In order to reach that, the many strains sharing the exact same sequence than the references strains ATCC66891...BMP0270 should be listed separately. The same can be done for strains identical with BMP3064.

We improved the Figure

Line 125: “the lowest level of rain and temperature were detected in 2014”: the temperature in 2015 was lower.

We referred to this sentence to the data from heading to harvest: we specified it in the new version

There are many typos that I cannot list here, but here were the main errors I found:

l. 35: stage result the -> stage is the

 CHANGED

l. 62: micotoxin -> mycotoxin

CHANGED

l. 64 I did not understand “while 63 ATX-II and ATX-III are both cellular transformation includes”

We changed the sentence

l. 65 results to be -> is

CHANGED

l. 131 morphological -> morphologically.

CHANGED

l. 144: as shown in Figure 1 -> as determined usng reference strains (Figure 1)

CHANGED

l. 147 “subclade   grouped” -> subclade A1 (?) grouped

We reported it in the text

L 167 and following: mg Kg-1 is not correct. You should use either mg.kg-1 or p.p.m.

CHANGED

l. 218 I did not understand “poses a great attention”

We changed the sentence

l. 226 I did not understand “solicited the need” 

We changed the sentence

l. 228-232 should be heavily reworded

we reworded as follows:

The samples analyzed in this study were highly contaminated  by fungal species belonging mainly to Alternaria, Cladosporium, Epicoccum, Fusarium, and Stemphylum genera (97 samples out of 100). The Alternaria occurrence was very high for incidence and frequency. Indeed, 80 out of the 100 samples analyzed, were contaminated by Alternaria at a value higher than 10% of the kernels.

l. 273: there is no “ref [378]” Corrected

l. 285: I did not understand “chemically unraveled”

CHANGED WITH “characterized”

l.324: “elaborated” -> “analysed” or “used”

CHANGED

Starting l. 338, there are may cases of missing spaces. I detected missing spaces in lines 338, 343, 349, 378, 398, 399, 400, 401, 402, 403.

CORRECTED

l. 375: Foe -> For

CHANGED

l. 391: surnatant -> supernatant

CHANGED

l. 395: AME appears in blue.

CORRECTED

Reviewer 2 Report

This is an interesting paper describing the Alternaria species found infecting wheat grown in Tuscany Italy.  The methods used are good, combining morphological and sequence analyses with mycotoxin analyses.  The problems with the paper are primarily in the presentation of results.  The paper needs moderate revision to fix English grammar, which are too numerous to list individually here. Several examples are given.

Line 17 change to  "...40 strains produced TA, and 26..."

Line 31  change 'attested on' to 'recorded at'

Line 47-49 change to  "...which causes a reduction in flour quality for the release of black points in the flour and in the resulting baked bread, and the loss of nutritional value with decreased strach, soluble carbohydrates and increased phenolics,..."

Line 88 and throughout,  change micotoxin to mycotoxin

Table 2 needs additional explanation of what the numbers in the second and third columns refer to.  Are these the number of samples tested?, the number of samples containing fungi?, the number of samples infected with Alternaria?

The quality of Figure 1 is not very good, with too small and blurry fonts.  Hopefully a legible copy will be acquired for publication.

Several places are missing information.  in Line 147, which subclade (A1?) included the 70 field strains?

In Line 173, which subclade (A1?) had 28 strains?

In Line 175,  need to add that Alternaria Section Clade A produced this mycotoxin,

LIne 64, missing the end of the sentence after 'includes'.

Under the Materials and Methods, 5.5 Phylogenetic analyses section, need to include a statement that section designations for Alternaria were made according to [38], since Woudenberg used different section designations than Lawrence.

Line 104, Consider adding a supplementary table that lists the other fungi and maybe the most predominant by year and location. This is especially important since it is mentioned again in the Discussion, lines 229-230.

Line 425 [8] Somma et al is not a valid reference since it has not been accepted for publication.  The information would have to be cited as unpublished.

Author Response

REVIEWER 2 REPORT

This is an interesting paper describing the Alternaria species found infecting wheat grown in Tuscany Italy. The methods used are good, combining morphological and sequence analyses with mycotoxin analyses. The problems with the paper are primarily in the presentation of results.  The paper needs moderate revision to fix English grammar, which are too numerous to list individually here. Several examples are given.

Line 17 change to "...40 strains produced TA, and 26..."

CHANGED

Line 31  change 'attested on' to 'recorded at'

CHANGED

Line 47-49 change to "...which causes a reduction in flour quality for the release of black points in the flour and in the resulting baked bread, and the loss of nutritional value with decreased strach, soluble carbohydrates and increased phenolics,..."

CHANGED

Line 88 and throughout,  change micotoxin to mycotoxin

CHANGED

Table 2 needs additional explanation of what the numbers in the second and third columns refer to.  Are these the number of samples tested?, the number of samples containing fungi?, the number of samples infected with Alternaria?

CHANGED, accordingly

The quality of Figure 1 is not very good, with too small and blurry fonts.  Hopefully a legible copy will be acquired for publication.

CHANGED accordingly

Several places are missing information. in Line 147, which subclade (A1?) included the 70 field strains?

We reported the information

In Line 173, which subclade (A1?) had 28 strains?

The information has been reported

In Line 175, need to add that Alternaria Section Clade A produced this mycotoxin,

CORRECTED

LIne 64, missing the end of the sentence after 'includes'.

We reworded the sentence

Under the Materials and Methods, 5.5 Phylogenetic analyses section, need to include a statement that section designations for Alternaria were made according to [38], since Woudenberg used different section designations than Lawrence.

CORRECTED

Line 104, Consider adding a supplementary table that lists the other fungi and maybe the most predominant by year and location. This is especially important since it is mentioned again in the Discussion, lines 229-230.

Table added

Line 425 [8] Somma et al is not a valid reference since it has not been accepted for publication. The information would have to be cited as unpublished.

We reported it as required by the journal
